# Willingness to Pay for Seasonal Influenza Vaccination among Children, Chronic Disease Patients, and the Elderly in China: A National Cross-Sectional Survey

**DOI:** 10.3390/vaccines8030405

**Published:** 2020-07-22

**Authors:** Xiaozhen Lai, Hongguo Rong, Xiaochen Ma, Zhiyuan Hou, Shunping Li, Rize Jing, Haijun Zhang, Zhibin Peng, Luzhao Feng, Hai Fang

**Affiliations:** 1School of Public Health, Peking University, Beijing 100083, China; laixiaozhen@pku.edu.cn (X.L.); rzjing2015@hsc.pku.edu.cn (R.J.); zhanghj966@bjmu.edu.cn (H.Z.); 2China Center for Health Development Studies, Peking University, Beijing 100083, China; hgrong@hsc.pku.edu.cn (H.R.); xma@hsc.pku.edu.cn (X.M.); 3Institute for Excellence in Evidence-Based Chinese Medicine, Beijing University of Chinese Medicine, Beijing 100029, China; 4School of Public Health, Fudan University, Shanghai 200032, China; zyhou@fudan.edu.cn; 5School of Health Care Management, Cheeloo College of Medicine, Shandong University, Shandong 250012, China; lishunping@sdu.edu.cn; 6NHC Key Laboratory of Health Economics and Policy Research (Shandong University), Shandong 250012, China; 7Division of Infectious Diseases, Chinese Center for Disease Control and Prevention, Beijing 102206, China; pengzb@chinacdc.cn; 8School of Population Medicine and Public Health, Chinese Academy of Medical Sciences & Peking Union Medical College, Beijing 100730, China; 9Peking University Health Science Center—Chinese Center for Disease Control and Prevention Joint Center for Vaccine Economics, Beijing 100083, China; 10Key Laboratory of Reproductive Health, National Health Commission of the People’s Republic of China, Beijing 100083, China

**Keywords:** influenza, vaccination, willingness to pay, priority group

## Abstract

*Background*: The disease burden of seasonal influenza is substantial in China, while the vaccination rate is extremely low, and most people have to pay 100% for vaccination. This study aims to examine willingness to pay (WTP) and recommended financing sources for influenza vaccination among children, chronic disease patients, and the elderly in China and determine feasible measures to expand vaccination coverage. *Methods*: From August to October 2019, 6668 children’s caregivers, 1735 chronic disease patients, and 3849 elderly people were recruited from 10 provinces in China. An on-site survey was conducted via a especially designed PAD system. Tobit regression was adopted to predict the influencing factors of WTP. *Results*: The average WTP was 127.5 yuan (USD18.0) for children, 96.5 yuan (USD13.7) for chronic disease patients, and 88.1 yuan (USD12.5) for the elderly. Most participants in the three groups thought government subsidies (94.8%, 95.8%, and 95.5%) or health insurance (94.3%, 95.3%, and 94.5%) should cover part of the cost, and nearly four-fifths (80.1%, 79.5%, and 76.8%) believed that individuals should also pay for part. Tobit regression showed that a higher perceived importance of vaccination, knowing about priority groups, and considering that individuals should co-pay were promoters of WTP, while considering price as a hindrance lowered WTP. *Conclusions*: The WTP for influenza vaccination among children, chronic disease patients, and the elderly in China is fairly high, suggesting that price is not the primary hindrance and there is room to expand immunization. Most participants expected the government and/or health insurance to pay part of the cost, and such supportive funding could act as a promotive policy “signal” to improve vaccine uptake. Influenza-related health education is also needed to expand vaccine coverage.

## 1. Introduction

As estimated by the World Health Organization (WHO) in 2018, annual seasonal influenza epidemics could result in about 3 to 5 million severe cases and about 290,000 to 650,000 respiratory deaths [1]. The disease burden of seasonal influenza is also substantial in China, causing about 88,000 influenza-associated respiratory deaths annually [2]. Immunization has been proven to be one of the most cost-effective health investments to prevent and control influenza [3]. However, the coverage rate of influenza vaccination in China has been extremely low in the past 15 years, with only 2% of the entire population being immunized, far lower than that in Western developed countries and some developing countries in Asia and South America [2].

The technical guidance for influenza vaccination in China (2019–2020) clearly points out that children (aged 6–59 months), patients with chronic diseases, and the elderly (aged 60 years or older) are among the priority groups for influenza vaccination, together with healthcare workers, family members and caregivers of infants under 6 months, and pregnant women [4]. However, at present, influenza vaccination has not been included in China’s National Immunization Program (NIP), and the expenses are paid totally out of pocket on most occasions, which is an important barrier to expanding vaccine coverage [5]. A previous study identified the level of influenza vaccination coverage for target groups in China and found vaccination rates of 26% among children younger than 5 years old, 7.4% among the elderly 60 years old or above, and 9.4% among chronic disease patients [6]. The vaccination rates for target groups were higher than that for the entire population (2%), but still far below previously reported rates in developed countries [7,8,9]. To solve the problem, some local governments with larger budget surplus in China, such as the Beijing and Shenzhen municipal governments, have attempted to provide influenza vaccination free of charge for schoolchildren and the elderly [10].

Until now, few studies have been published on willingness to pay (WTP) and financing strategies for influenza vaccination among children, patients with chronic diseases, and the elderly in China, even though they are priority groups recommended by the WHO [11]. The present study aims to examine WTP and recommended financing sources for seasonal influenza vaccination among children aged 6–59 months, patients with chronic diseases 18–59 years old, and elderly people older than 60 years, in order to determine feasible measures to improve the coverage rate for these three priority groups in China. It was hypothesized that the reported WTP would be higher among children than the other groups and many respondents would recommend the government and/or health insurance as a financing source for influenza vaccination.

## 2. Methods 

### 2.1. Study Population

From August to October 2019, a total of 148 community health centers from 10 provinces in China were approached to join the national survey on influenza vaccination WTP among three priority groups: children aged 6–59 months, chronic disease patients aged 18–59 years, and the elderly aged above 60 years (Financing Strategies of Influenza Vaccination in China, NCT04038333). For children aged 6–59 months, we asked their parents or grandparents who accompanied them to health centers to finish the immunization procedure. In China, it is compulsory for children under 5 years old to go through routine immunization schedules in health centers [4], so we could collect an unbiased sample in this way. For chronic disease patients and the elderly, we asked them in health centers or gathered them in neighborhood committees. Selection bias would happen if we only interviewed the elderly in health centers when they were visiting doctors, so we also approached neighborhood committees to reduce the bias among the elderly. This study was ethically reviewed and approved by the Peking University Institutional Review Board (IRB00001052-19076), and written informed consent was obtained from individual or guardian participants.

We calculated the sample size under the assumption that the predicted influenza vaccine coverage would be 30% among children, 10% among the elderly, and 10% among those with chronic diseases. With an allowable error of 5%, a sample size of 323, 138, and 138 was determined for the three groups in each province. To allow for disqualification of incomplete questionnaires, we increased the sample size by 10%, with a final target sample population of 355, 152, and 152 in each province. In practice, we collected a larger sample size than expected to increase the reliability of the results.

A multistage sampling method was adopted in this survey. First, ten provinces/municipalities were selected based on the Division of Central and Local Financial Governance and Expenditure Responsibilities in the Healthcare Sector released by the State Council in 2018, which stratifies the 31 provinces/municipalities in mainland China into five layers [12]. The division of expenditure responsibility between central and local governments differed across layers, with ratios of 8:2 (first layer), 6:4, 5:5, 3:7, and 1:9 (fifth layer). In terms of location, socioeconomic development, and accessibility, 10 provinces/municipalities (3, 3, 1, 1, and 2 in each layer) were chosen, with their location and 2018 per capita GDP rank (e.g., 1/31), as shown in Figure 1. Second, in each province/municipality, a capital city or well-developed district (in municipalities) and a non-capital city or less-developed district were selected. Third, two subdistricts/counties were chosen in each city or district, in which three or more immunization centers (in community health centers or township clinics) and the corresponding neighborhood committees were approached to participate in the survey.

### 2.2. Measures

The on-site survey was conducted by trained interviewers using a specially designed online questionnaire system in portable android device (PAD). Automatic logical proofreading was adopted to reduce input errors and missing values. In addition, interview recordings were uploaded and spot-checked by quality control personnel to find and correct problems in time. The online questionnaire was divided into four parts: (1) sociodemographics including age, gender, education level, household monthly per capita income, place of residence (urban or rural), self-reported health status, etc.; (2) WTP for influenza vaccination; (3) recommended financing sources of influenza vaccination including individuals, medical insurance, and the government; and (4) knowledge and perception of influenza and influenza vaccination, including perceived possibility of catching and perceived severity of influenza; perceived importance, safety, and efficacy of influenza vaccination; knowledge of the priority groups; perceived hindrances to vaccination (price, distance, and time); trust in doctors’ vaccination advice; and doubts about influenza vaccination. In the original questionnaire, self-reported health status and some variables concerning knowledge and perception of influenza and influenza vaccination were designed as 5-point Likert scales. To simplify the analysis, we regrouped the answers very high and high as high, and fair, low and very low as low to construct binary variables.

Willingness to pay (WTP) refers to the largest sum of money an individual will agree to pay for a product or service [13]. The contingent valuation method (CVM) is widely used to obtain WTP, usually by means of questionnaire surveys [14]. Specific CVMs include bidding games, payment cards, dichotomous choice, etc. To compare, although a step-by-step bidding game is more time-consuming, its results are more precise due to narrower price deviation and more time left for respondents by asking repeated questions [15]. This study measured WTP using a modified step-by-step bidding game. For each participant, a starting point ranging from 0 to 150 yuan (1 yuan = USD0.1415 on 3 July 2020) was randomly presented by the questionnaire system, as 150 yuan was considered to be the price ceiling for both trivalent and quadrivalent influenza vaccines in China. The bidding process was terminated after the interviewer’s WTP or endpoint (0 or 150 Yuan) was reached. In addition, it was indicated that a reported WTP of zero does not necessarily indicate a true WTP of zero, but rather some respondents protesting against paying for the service [16]. To distinguish those reporting “false” zeros, we further separated participants who refused to be vaccinated against influenza even if it was free in order to examine the WTP and recommended financing sources of those who did not resist vaccination.

After measuring WTP using the bidding game, we asked the participants what percentage they were willing to pay for influenza vaccination assuming the total cost was 50 yuan (which approximates the most common price of trivalent influenza vaccines in China), and whether they were willing to pay the total cost of 50 yuan to further examine their WTP. As for recommended financing sources, we asked the participants three questions “do you think individuals/medical insurance/the government should participate in the payment of influenza vaccination for you/your child?” and collected their answers.

### 2.3. Statistical Analysis

Continuous variables are described as mean (standard error), and discrete variables are shown as percentages (%). The chi-square and Mann–Whitney tests were used to assess differences in sample characteristics. Detailed distribution of WTP or self-payment ratio was conceptualized as cumulative frequency line graphs, which take the proportion of participants as the horizontal axis and WTP amount or payment ratio as the vertical axis. Multivariate tobit regression was adopted to predict the influencing factors of WTP (concentrating on boundary values 0–150), and results are shown as coefficient (standard error). A two-sided *p*-value below 0.05 was considered statistically significant. All data were analyzed using Stata version 14.0 (Stata Corp., College Station, TX, USA).

In the tobit regression, the “3Cs” model of vaccine hesitancy was adopted to identify influencing factors, which involves three factors: complacency (recognition of the need, value, and importance of vaccines), convenience (vaccine accessibility), and confidence (degree of trust in vaccines) [17]. This study constructed the regression according to the 3Cs model, in which the dependent variable was WTP value measured in the step-by-step bidding game, and independent variables included sociodemographics and the 3Cs variables. More specifically, the complacency group included perceived possibility of catching influenza, perceived severity of influenza, perceived importance of influenza vaccination, knowledge of priority groups (children, chronic disease patients, and the elderly), and attitude toward individual participation in payment; the convenience group included hindrances to vaccination (price, distance, and time) and shortage of vaccines in clinics; and the confidence group included perceived safety or efficacy of influenza vaccination, trust in doctors’ vaccination advice, and doubts about vaccination.

## 3. Results

### 3.1. Study Sample Characteristics 

A total of 12,252 valid questionnaires (6668 for children, 1735 for patients with chronic diseases, and 3849 for the elderly) were received, with an effective response rate of 99.80%. Table 1 shows the sociodemographics and perception and knowledge of influenza vaccination among the three groups of study participants. There were significant differences across the three groups in terms of comparable characteristics (all *p* < 0.01), except for the shortage of influenza vaccines in clinics (*p* = 0.07). Generally speaking, the education and household income levels of children’s caregivers were higher than those of the other two groups. Among the children, four age groups, <1, 1, 2, and 3–5 years old, accounted for 26.5%, 29.4%, 18.6%, and 25.5%, respectively. There were slightly more boys than girls (52.4% vs. 47.6%). The majority of caregivers were parents (82.9%), and 76.3% of them were younger than 40 years old. In the other two groups, patients with chronic diseases 50–59 years of age accounted for 77.8%, and the elderly 60–79 years of age accounted for 91.9%. More women participated in the survey in both groups, and most of them were married. We further compared the distribution of gender, age, and marital status in the three groups with those recorded in China Population and Employment Statistics Yearbook 2019 [18] and found similar results, indicating the national representativeness of the population collected in this study.

The majority of participants reported a low possibility of catching influenza (54.2%, 69.0%, and 73.2%), high severity of influenza (74.7%, 67.4%, and 65.2%), and high importance of influenza vaccination (90.0%, 78.8%, and 73.2%). More caregivers of children (57.5%) knew that their children were among the priority groups for influenza vaccination, while the majority of chronic disease patients (60.4%) and elderly (60.9%) did not know that they were among the priority groups. Many participants did not think that price (82.6%, 63.8%, and 64.8%), distance, and time (87.4%, 81.2%, and 85.4%) were hindrances to vaccination. In the past year, nearly half of the participants (47.5%, 45.5%, and 45.4%) experienced a shortage of influenza vaccines at the clinics they usually went to. A majority of participants considered influenza vaccines to be safe (80.6%, 72.2%, and 67.2%) and efficacious (79.7%, 72.5%, and 65.5%), and most of them trusted doctors’ vaccination advice (90.8%, 86.5%, and 83.3%). For caregivers of children, a majority had doubts about vaccination (60.2%) or were willing to vaccinate but had not done it (51.4%), which was different from chronic disease patients and the elderly.

### 3.2. Willingness to Pay

Figure 2 shows the distribution of WTP among the three groups. The median WTP for influenza vaccination among children aged 6–59 months, patients with chronic diseases aged 18–59 years, and the elderly aged 60 and above was about 150 yuan, 95 yuan, and 80 yuan, respectively. Nearly two-thirds of children’s caregivers and one-third of chronic disease patients and the elderly were willing to pay a maximum of 150 yuan for influenza vaccination. In particular, the WTP of 5.55% of caregivers, 16.9% of chronic disease patients, and 25.7% of the elderly was 0 yuan. We further separated those who refused to be vaccinated against influenza even if it was free to distinguish those reporting “false” zeros. A total of 144 caregivers (2.16%), 132 chronic disease patients (7.61%), and 510 elderly people (13.3%) were identified to have “false” zeros (recorded as −100 yuan in Figure 2), and thus were excluded from the following analysis.

Table 2 further summarizes the willingness to pay for seasonal influenza vaccination among the surveyed population after excluding those with “false” zeros. The average WTP was 127.5 yuan for caregivers of children, 96.5 yuan for chronic disease patients, and 88.1 yuan for the elderly. The participants were further asked what percentage they were willing to pay for influenza vaccination assuming the total cost was 50 yuan. It was found that the three groups were willing to pay 51.7%, 43.7%, and 33.6% of the total cost if there was co-funding. Moreover, even if they were required to pay fully for influenza vaccination (50 yuan), there were still 92.8%, 75.4%, and 70.4% participants willing to pay for their children or themselves.

Figure 3 displays the distribution of expected self-payment ratios for influenza vaccination among the three groups in detail after excluding the “false” zeros. As described above, the three groups were willing to pay, on average, 51.7%, 43.7%, and 33.6% of the total cost of 50 yuan. The figure also shows that nearly one-fifth of the respondents in three priority groups were willing to pay 100% for influenza vaccination, while 8–18% of the respondents were reluctant to partly pay for it.

### 3.3. Recommended Financing Sources

Table 3 displays recommended financing sources of seasonal influenza vaccination among the surveyed population after excluding those with “false” zeros. The majority of participants believed that it should be jointly paid by individuals, health insurance, and governments. Specifically, most respondents considered that the government (94.8%, 95.8%, and 95.5%) and health insurance (94.3%, 95.3%, and 94.5%) should partly pay. In addition, nearly four-fifths of respondents (80.1%, 79.5%, and 76.8%) believed that individuals should also participate in paying for influenza vaccination, which reflects the idea that individuals are responsible for their own health. 

### 3.4. Influencing Factors of WTP

Table 4 shows the tobit regression after excluding “false” zeros (see Appendix A for the results before exclusion), and results are shown as coefficient (standard error) (see Appendix B for marginal effects). For the caregivers of children, the multivariate regression was adjusted for sociodemographics including children’s age and gender, respondents’ information (age, family relation, education level, basic medical insurance type), household monthly per capita income, place of residence, self-reported health status, influenza-like illness in the past year, and province. The results show that higher perceived severity of influenza (Coef = 10.87, *p* < 0.05), higher perceived importance of influenza vaccination (Coef = 37.73, *p* < 0.05), knowing that children are a priority group for influenza vaccination (Coef = 10.41, *p* < 0.05), and considering that individuals should participate in payment (Coef = 47.02, *p* < 0.05) would increase their WTP. Additionally, caregivers had lower WTP if the vaccine price was considered as a hindrance (Coef = −67.50, *p* < 0.05). Higher WTP was also related to fewer doubts about vaccination (Coef = −6.64, *p* < 0.05) and being willing to vaccinate but have not done so (Coef = 16.64, *p* < 0.05).

The regressions for chronic disease patients and the elderly were controlled for age, gender, marital status, education level, basic medical insurance type, household monthly per capita income, place of residence, self-reported health status, influenza-like illness in the past year, and province. For patients with chronic diseases, higher perceived importance of influenza vaccination (Coef = 28.84, *p* < 0.05), knowing that they are a priority groups for influenza vaccination (Coef = 12.76, *p* < 0.05), and considering that individuals should participate in payment (Coef = 53.49, *p* < 0.05) would increase their WTP. Moreover, the feeling that price hindered vaccination behavior would lower their WTP (Coef = −57.63, *p* < 0.05), while the having trust in doctors’ vaccination advice would increase their WTP (Coef = 14.73, *p* < 0.05). For the elderly, higher perceived possibility of catching influenza (Coef = 11.32, *p* < 0.05), higher perceived importance of influenza vaccination (Coef = 20.12, *p* < 0.05), knowing that they are a priority group for influenza vaccination (Coef = 12.73, *p* < 0.05), and considering that individuals should participate in payment (Coef = 71.70, *p* < 0.05) would increase their WTP. Besides, the elderly would lower their WTP if they considered that price was a hindrance (Coef = −56.68, *p* < 0.05). Higher WTP also resulted from more trust in doctors’ vaccination advice (Coef = 18.90, *p* < 0.1), fewer doubts about vaccination (Coef = −10.16, *p* < 0.05), and being willing to vaccinate but not having done so (Coef = 11.70, *p* < 0.05).

## 4. Discussion

To the best of our knowledge, this is the first study to use a nationally representative sample from 10 provinces in China to investigate WTP for influenza immunization among three priority groups: children, patients with chronic diseases, and the elderly. In accordance with the hypotheses, this study shows that the WTP for influenza vaccination was fairly high, children’s caregivers had higher WTP (127.5 yuan) than patients with chronic diseases and the elderly (96.5 and 88.1 yuan, respectively), and most participants in the three groups thought government subsidies (94.8%, 95.8%, and 95.5%) or health insurance (94.3%, 95.3%, and 94.5%) should cover part of the cost.

As introduced above, currently influenza vaccination is not included in China’s NIP, and the expense is paid totally out of pocket on most occasions. Influenza vaccines provided in different regions differ in price due to varied costs of procurement, transportation, and labor. In the present study, 150 yuan was set as the upper limit, as it was considered to be the price ceiling for influenza vaccines in China. Generally speaking, a trivalent influenza vaccine (including the service fee) costs about 40–90 yuan. The quadrivalent vaccine is more expensive, costing about 140 yuan, but only a small number of people are vaccinated with it since it was recently made available in China, and the cost is expected to go down after years of use [19]. As Table 2 indicates, if participants were required to fully pay for influenza vaccination at the price of 50 yuan, 92.8% of children’s caregivers, 75.4% of chronic disease patients, and 70.4% of elderly people were willing to pay for it. According to Figure 2, the higher the assumed price, the fewer participants would be willing to pay for it.

A study conducted in 2013 used a traditional bidding game method to interview households with children aged 0–3 years in three provinces in China, and found that the median WTP for influenza vaccination was 60 yuan [20], lower than the mean WTP reported in the present study (127.5 yuan for children aged 6–59 months). Possible explanations for the gap lie in not only time and regional differences of study design, but also the method to elicit WTP. In a traditional bidding game, the investigator sets two starting points in advance and randomly assigns 50% of the samples to start from the lower point and the other 50% from the higher point, while different starting points may affect the respondent’s choice, causing “starting point bias” [15]. In this study, WTP was measured using a modified step-by-step bidding game. For each participant, a starting point ranging from 0 to 150 yuan was randomly given by the questionnaire system. In this way, the impact of “bipolar” starting points can be avoided to achieve more reliable results.

Previous overseas studies have shown that the price of influenza vaccine is an important barrier to vaccine coverage [21,22]. However, this study shows that WTP for influenza vaccination is fairly high, indicating that price may not be a large hindrance to improving the uptake rate of influenza vaccination, which remains low in China. Possible explanations for the gap between the low vaccination rate and high WTP include: (1) some respondents have never heard of influenza vaccination or its priority groups before the survey, but they had high WTP after being exposed to the knowledge; (2) WTP is a choice made under hypothetical conditions, while immunization behavior is affected by many other factors, such as distance and time constraints [20,22]; (3) some interviewees may overstate their WTP to stress on the importance they attach to health; and (4) participants with high WTP may change their vaccination choice due to risk aversion if they feel there are potential safety problems or it is not necessary to get immunized [23].

Nearly 95% of respondents considered that the government and health insurance should participate in the payment of influenza vaccination, reminding us that the participation of a third party, i.e., the government or health insurance, could accelerate the expansion of vaccine coverage. A study conducted in the United States found that the Medicaid reimbursement rate was closely related to the uptake rate of influenza vaccination among children [24]. A study in Ontario, Canada, also pointed out that support for government spending would significantly promote the decision to vaccinate [25]. In Japan, it was found that increasing the subsidy amount by 1000 yen (USD10) led to a one percentage point increase in the vaccination rate among the elderly, thus improving their health outcomes [26]. In China, a cost-effectiveness study estimated that the threshold vaccination cost is 71.48 yuan (USD10.19) to achieve a fully funded vaccination program for older adults aged ≥60 years [27]. In the future, combining the financing sources for influenza vaccination (individuals, health insurance, and government) may become a promotive policy signal for individuals to vaccinate against influenza. Besides, vaccination is widely recognized as an action with strong positive externalities [28,29], so public intervention is expected to drive vaccine coverage to a socially optimal level [30]. At present, a few areas with larger budget surplus in China are trying to provide government financial subsidies and medical insurance compensation for influenza vaccination, but most of the subsidies and compensation policies are designed for the elderly and school-age children. Caregivers of children 5 to 59 months of age and patients with chronic diseases under 60 years old still have to pay the full cost themselves [10].

As for the influencing factors of WTP among the three groups, we find that higher perceived importance of influenza vaccination, knowing the priority groups, and considering that individuals should participate in payment were shared promoters of WTP, while considering price as a hindrance would lower participants’ WTP. By contrast, WTP did not significantly change whether participants considered distance and time as hindrances, whether there used to be a vaccine shortage in clinics, or whether they perceived high safety or high efficacy of influenza vaccination. The regression results suggest that efforts should be made to promote influenza-related health education to the public on the susceptibility and severity of influenza, the importance of vaccination, the priority groups for immunization, etc., so as to help people correctly understand influenza and influenza vaccination.

The present study also has a few limitations. First, some of the elderly people surveyed in this study were recruited from community health centers while they were seeking primary medical services, which may have resulted in selection bias. Nevertheless, given the high prevalence of noncommunicable diseases among Chinese elderly [31] and the fact that community health centers mainly provide primary health care, the bias can be reduced. Second, self-reported responses may be subject to recall bias and a tendency to report socially desirable responses; therefore, the results should be interpreted with caution. Third, the cross-sectional design used in this study does not allow for causal conclusions, so causality cannot be inferred with certainty. Despite these limitations, the nationally representative sample was large, with a diverse sociodemographic population, thus offering good generalizability for the three priority groups in China.

## 5. Conclusions

In conclusion, the willingness to pay for influenza vaccination for children, chronic disease patients, and the elderly in China is fairly high, suggesting that price may not be the primary hindrance, and there is a great opportunity to immunize more people in need. While individuals can pay part of the immunization fees by themselves as the first person responsible for their health, government subsidies and health insurance are also expected to cover part of the cost instead of leaving the individual to pay out of pocket only. Such a supportive measure taken by the government and/or health insurance could act as a promotive policy signal to improve the vaccine uptake rate. In addition, public influenza-related health education is needed in future health promotion practices to expand the coverage of influenza vaccination.

## Figures and Tables

**Figure 1 vaccines-08-00405-f001:**
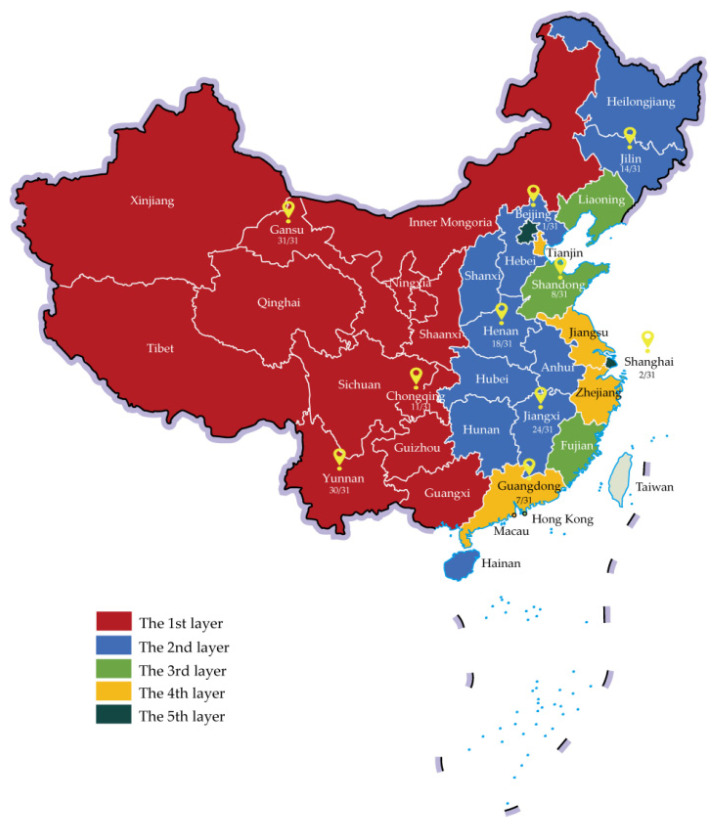
Ten provinces/municipalities selected for survey on willingness to pay for seasonal influenza vaccination in China, 2019.

**Figure 2 vaccines-08-00405-f002:**
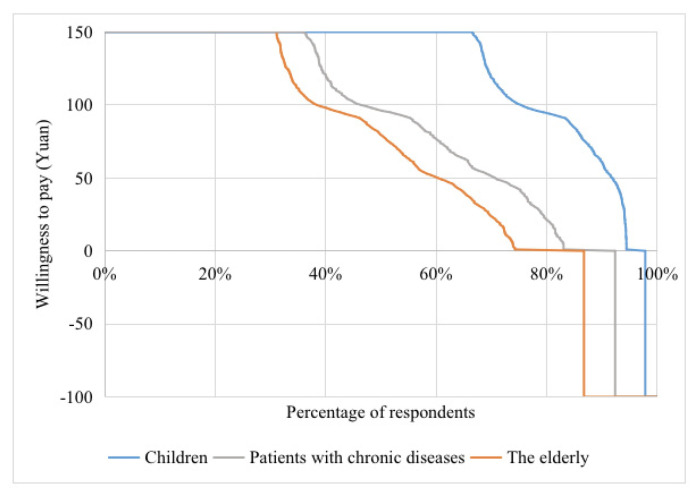
Distribution of willingness to pay for influenza vaccination among three priority groups in China, 2019.

**Figure 3 vaccines-08-00405-f003:**
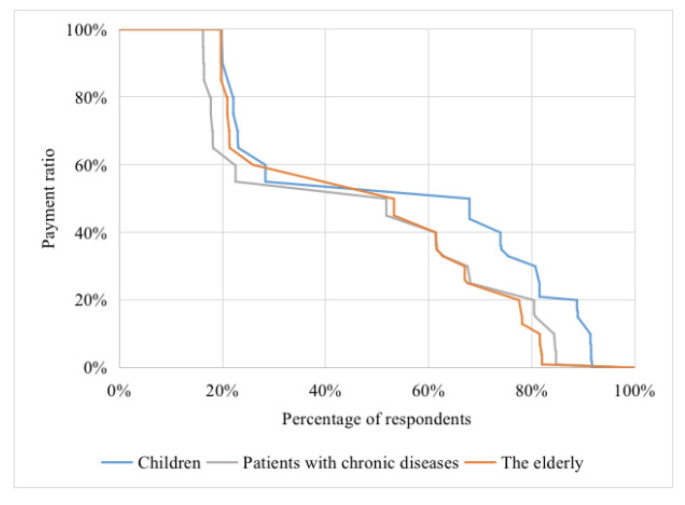
Distribution of expected self-payment ratios of influenza vaccination among three priority groups.

**Table 1 vaccines-08-00405-t001:** Characteristics of 12,252 participants included in an analysis of willingness to pay for seasonal influenza vaccination in China, 2019.

Characteristics	Children Aged 6–59 Months, n (%)	Chronic Disease Patients Aged 18–59 Years, n (%)	Elderly Aged above 60 Years, n (%)	*p*-Value
**Sociodemographics**				
Age (years)				–
<1	1768 (26.5)	–	–	–
1	1959 (29.4)	–	–	–
2	1241 (18.6)	–	–	–
3–5	1700 (25.5)	–	–	–
18–39	–	72 (4.1)	–	–
40–49	–	313 (18.0)	–	–
50–59	–	1350 (77.9)	–	–
60–69	–	–	2045 (53.1)	–
70–79	–	–	1491 (38.7)	–
≥80	–	–	313 (8.2)	–
Age of respondent (years)				–
<30	2049 (30.7)	–	–	–
30–39	3039 (45.6)	–	–	–
40–49	569 (8.5)	–	–	–
≥50	1011 (15.2)	–	–	–
Relationship between respondent and child				–
Father	1122 (16.8)	–	–	–
Mother	4405 (66.1)	–	–	–
Grandfather	213 (3.2)	–	–	–
Grandmother	928 (13.9)	–	–	–
Gender				<0.01 *
Male	3497 (52.4)	596 (34.4)	1515 (39.4)	–
Female	3171 (47.6)	1139 (65.6)	2334 (60.6)	–
Marital status				<0.01 *
Married	–	1575 (90.8)	2909 (75.6)	–
Unmarried/divorced/widowed	–	160 (9.2)	940 (24.4)	–
Education level ^a^				<0.01 *
Elementary school and below	674 (10.1)	618 (35.6)	2146 (55.8)	–
Junior high school	1710 (25.6)	580 (33.4)	915 (23.8)	–
High school/vocational school	1503 (22.5)	361 (20.8)	583 (15.1)	–
Junior college	1250 (18.7)	108 (6.2)	125 (3.3)	–
Four-year college and above	1531 (23.1)	68 (4.0)	80 (2.0)	–
Household monthly per capita income (thousand yuan) ^b,c^	2.7 (2.8)	1.8 (2.4)	1.7 (1.6)	<0.01 *
Place of residence				<0.01 *
Urban	3854 (57.8)	904 (52.1)	2107 (54.7)	–
Rural	2814 (42.2)	831 (47.9)	1742 (45.3)	–
Basic medical insurance type ^a^				<0.01 *
Medical insurance for urban and rural residents	4081 (61.2)	1212 (69.9)	2551 (66.3)	–
Urban employee medical insurance	2380 (35.7)	492 (28.4)	1189 (30.9)	–
Without basic medical insurance	207 (3.1)	31 (1.7)	109 (2.8)	–
Self-reported health status				<0.01 *
Good	5812 (87.2)	596 (34.4)	1578 (41.0)	–
Fair or poor	856 (12.8)	1139 (65.6)	2271 (59.0)	–
Had influenza-like illness in the past year				<0.01 *
Yes	3049 (45.7)	291 (16.8)	489 (12.7)	–
No	3619 (54.3)	1444 (83.2)	3360 (87.3)	–
Province				<0.01 *
Beijing	668 (10.0)	153 (8.8)	332 (8.6)	–
Shanghai	602 (9.0)	158 (9.1)	332 (8.6)	–
Jilin	681 (10.2)	170 (9.8)	391 (10.2)	–
Yunnan	629 (9.5)	155 (8.9)	308 (8.0)	–
Shandong	626 (9.4)	172 (9.9)	355 (9.2)	–
Guangdong	621 (9.3)	153 (8.8)	334 (8.7)	–
Jiangxi	621 (9.3)	165 (9.5)	330 (8.6)	–
Gansu	742 (11.1)	225 (13.0)	577 (15.0)	–
Chongqing	848 (12.7)	222 (12.8)	576 (15.0)	–
Henan	630 (9.5)	162 (9.4)	314 (8.1)	–
**Knowledge and perception of influenza and vaccination**				
Perceived high possibility of catching influenza				<0.01 *
Yes	3056 (45.8)	537 (31.0)	1032 (26.8)	–
No	3612 (54.2)	1198 (69.0)	2817 (73.2)	–
Perceived high severity of influenza				<0.01 *
Yes	4982 (74.7)	1170 (67.4)	2508 (65.2)	–
No	1686 (25.3)	565 (32.6)	1341 (34.8)	–
Perceived high importance of influenza vaccination				<0.01 *
Yes	6000 (90.0)	1367 (78.8)	2819 (73.2)	–
No	668 (10.0)	368 (21.2)	1030 (26.8)	–
Knowledge of priority groups (children, chronic disease patients, elderly)				<0.01 *
Yes	3834 (57.5)	687 (39.6)	1505 (39.1)	–
No	2834 (42.5)	1048 (60.4)	2344 (60.9)	–
Price hinders vaccination				<0.01 *
Yes	1159 (17.4)	628 (36.2)	1353 (35.2)	–
No	5509 (82.6)	1107 (63.8)	2496 (64.8)	–
Distance and time hinder vaccination				<0.01 *
Yes	838 (12.6)	326 (18.8)	562 (14.6)	–
No	5830 (87.4)	1409 (81.2)	3287 (85.4)	–
Shortage of influenza vaccines in clinics				0.07
Yes	3165 (47.5)	789 (45.5)	1746 (45.4)	–
No	3503 (52.5)	946 (54.5)	2103 (54.6)	–
Perceived high safety of influenza vaccination				<0.01 *
Yes	5376 (80.6)	1252 (72.2)	2587 (67.2)	–
No	1292 (19.4)	483 (27.8)	1262 (32.8)	–
Perceived high efficacy of influenza vaccination				<0.01 *
Yes	5315 (79.7)	1258 (72.5)	2522 (65.5)	–
No	1353 (20.3)	477 (27.5)	1327 (34.5)	–
Trust in doctors’ vaccination advice				<0.01 *
Yes	6057 (90.8)	1500 (86.5)	3208 (83.3)	–
No	611 (9.2)	235 (13.5)	641 (16.7)	–
Doubts about vaccination				<0.01*
Yes	4014 (60.2)	624 (36.0)	1000 (26.0)	–
No	2654 (39.8)	1111 (64.0)	2849 (74.0)	–
Willing to vaccinate but have not done it				<0.01 *
Yes	3425 (51.4)	567 (32.7)	1071 (27.8)	–
No	3243 (48.6)	1168 (67.3)	2778 (72.2)	–

* Significant at the 5% level. ^a^ Education level and basic medical insurance type in column refer to adult respondents. ^b^ Shown as mean (SD).^c^ 1 yuan = USD0.1415 on 3 July 2020.

**Table 2 vaccines-08-00405-t002:** Willingness to pay for seasonal influenza vaccination.

Willingness to Pay	Children	Chronic Disease Patients	Elderly
Willingness to pay (yuan) ^a,b^	127.5 (39.2)	96.5 (53.7)	88.1 (56.8)
Expected self-payment ratio assuming total cost is 50 yuan (%) ^a,b^	51.7 (29.7)	43.7 (31.4)	44.9 (33.6)
Willingness to pay for total cost assuming it is 50 yuan, n (%) ^b^	6052 (92.8)	1208 (75.4)	2349 (70.4)

^a^ Shown as mean (SD). ^b^ 1 yuan = USD0.1415 on 3 July 2020.

**Table 3 vaccines-08-00405-t003:** Recommended financing sources for seasonal influenza vaccination.

Financing Sources	Children	Chronic Disease Patients	Elderly
Should individuals, medical insurance, or the government pay for influenza vaccination?			
Individuals, n (%)	5285 (81.0)	1275 (79.5)	2565 (76.8)
Medical insurance, n (%)	6152 (94.3)	1528 (95.3)	3155 (94.5)
Government, n (%)	6187 (94.8)	1536 (95.8)	3189 (95.5)
Total	6524	1603	3339

**Table 4 vaccines-08-00405-t004:** Tobit regression of willingness to pay for the three groups.

Factors		Children	Chronic Disease Patients	Elderly
Complacency	Perceived high possibility of catching influenza	−2.09 (3.14)	10.12 (5.17)	11.32 * (3.97)
	Perceived high severity of influenza	10.87 * (3.54)	−4.58 (5.22)	1.56 (3.85)
	Perceived high importance of influenza vaccination	37.73 * (5.08)	28.84 * (6.35)	20.12 * (4.59)
	Knowledge of priority groups for influenza vaccination (children, chronic disease patients, elderly)	10.41 * (3.14)	12.76 * (4.88)	12.73 * (3.52)
	Considering that individuals should participate in payment	47.02 * (3.65)	53.49 * (5.63)	71.70 * (4.14)
Convenience	Price hinders vaccination behavior	−67.50 * (3.81)	−57.63 * (4.97)	−56.68 * (3.73)
	Distance and time hinder vaccination behavior	−5.34 (4.37)	−7.35 (5.70)	−7.73 (4.70)
	Shortage of influenza vaccines in clinics	−2.49 (3.42)	5.34 (5.18)	5.85 (3.93)
Confidence	Perceived high safety of influenza vaccination	7.00 (4.14)	−0.89 (5.82)	7.03 (4.37)
	Perceived high efficacy of influenza vaccination	0.41 (4.02)	−0.65 (5.82)	3.81 (4.31)
	Trust in doctors’ vaccination advice	3.94 (5.22)	14.73 * (7.19)	18.90 * (4.93)
	Doubts about vaccination	−6.64 * (3.19)	−0.68 (4.79)	−10.16 * (4.00)
	Willing to vaccinate but have not done it	16.64 * (3.01)	6.49 (4.86)	11.70 * (3.70)

Notes: 1. Regression results are shown as coefficient (standard error). * Significant at the 5% level. 2. The regression for children was controlled for children‘s age and, gender, respondents’ information (age, family relation, education level, basic medical insurance type), household monthly per capita income, place of residence, self-reported health status, influenza-like illness in the past year, and province. 3. The regression for chronic disease patients and the elderly was controlled for age, gender, marital status, education level, basic medical insurance type, household monthly per capita income, place of residence, self-reported health status, influenza-like illness in the past year, and province.

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
