# Peer review of "Willingness to Pay for Seasonal Influenza Vaccination among Children, Chronic Disease Patients, and the Elderly in China: A National Cross-Sectional Survey"

_vaccines, 2020, doi:10.3390/vaccines8030405_

Round 1
Reviewer 1 Report
This paper reports the willingness to pay (WTP) of a large representative sample of over 10,000 Chinese people (children via their guardians, people with chronic diseases, elderly) from an on-site survey. The authors found that the WTP was highest among guardians of children (aged 6 months to 5 years), and lowest in the elderly. Most respondents believed that individuals should co-pay at least some part of the price of vaccination. Overall, WTP was nonetheless high, suggesting that price is not the major barrier to vaccination in China, where the uptake rates reported by the authors (if accurate) are indeed startling.
The paper is interesting, and investigates an important issue – what is it exactly that prevents people from getting vaccinated? How much does the price contribute to vaccine behaviours? In this regard the results are interesting, as it would appear that people are overall very willing to pay. Unsurprisingly, the respondents who reported the highest perceived risk of getting the flu, the highest perceived severity of the flu, the highest level of knowledge of target groups and the highest level of trust in doctors’ vaccination advice, were also the ones who had the highest WTP (namely, parents or guardians of small children). The authors fail to explore the distinction between WTP for others (i.e. for one’s children) and for oneself (chronic patients and elderly), but this is a somewhat philosophical debate that is likely beyond the scope of the present paper.
I have a few minor remarks for the authors’ consideration:
- The equivalence between Yuan and US dollars is given quite late in the paper; it would be better placed at the first mention of a price in Yuan. In the same vein, Table 1 mentions monthly income, per increase of one thousand– but per thousand what? What is the currency or unit? I suppose it is Yuan, but it would be helpful to specify (with equivalence).
- In Table 1, the authors indicate the self-reported health status of the respondents – could you indicate what instrument was used to report health status? The socio-demographic and health variables recorded could be listed in greater detail in the methods section, for example.
- The footnote to Table 3 indicates that *, ** and *** denote significance at the 10, 5 and 1 percent levels, respectively, but 10% is not usually considered statistically significant. A p-value of 0.05 corresponds to an alpha risk of 5%, anything more than that (or >0.05) is not statistically significant by convention. Please comment or correct. If this is a specificity of the statistical methods used, please clarify and explain.
- In the references, a review paper might be more appropriate as a reference for the contingent valuation method (e.g. DOI: 10.1016/s0168-8510(99)00010-x), rather than examples of use in individual countries (e.g. refs 11-13).
- Generally speaking, the English warrants thorough revision – there are a number of grammatical mistakes throughout the paper. For example, the authors use the phrase “almost out of pocket” on two different occasions, but there is obviously something missing here – either the expenses are “out-of-pocket”, or they are not. They cannot be “almost” out-of-pocket. There is likely a word missing (e.g. “almost totally out-of-pocket”). Page 11 of 13, line 283 – please correct “serenity” to “severity”. These are only two examples; there are many other parts that warrant correction.
Author Response
Thanks for Reviewer 1 for giving us an opportunity to revise the manuscript.
Responses to Reviewer 1
We would like to express our sincere gratitude to Reviewer 1 for the positive comments and constructive suggestions. We have carefully revised the manuscript according to the valuable suggestions. The revisions we have made are traced in the revised manuscript. Please find the following detailed point-by-point responses to your comments and suggestions.
- The equivalence between Yuan and US dollars is given quite late in the paper; it would be better placed at the first mention of a price in Yuan. In the same vein, Table 1 mentions monthly income, per increase of one thousand– but per thousand what? What is the currency or unit? I suppose it is Yuan, but it would be helpful to specify (with equivalence).
Response 1:
We would like to thank Reviewer 1 for this helpful comment on our manuscript.
1.1. As Reviewer 1 pointed out, the equivalence between Yuan and US dollars is given quite late in the original manuscript. We have relocated it at the first mention of a price in “Yuan”. In addition, we have also added the USD values of WTP in abstract to make convenient comparisons at the international level. Please see line 35-36 and 142-143 in the revised manuscript.
1.2. Reviewer 1’s understanding is correct. The monthly income mentioned in table 1 used the currency “Yuan”. We have specified the currency in the table, and add the equivalence between Yuan and US dollars as a note in every table which mentioned “Yuan”. Please see table 1 and table 2 in the revised manuscript.
- In Table 1, the authors indicate the self-reported health status of the respondents – could you indicate what instrument was used to report health status? The socio-demographic and health variables recorded could be listed in greater detail in the methods section, for example.
Response 2: Thanks for your kind suggestion. We further elaborated on variable measures used in this study. The online questionnaire can be divided into four parts: (1) sociodemographics including age, gender, education level, household monthly per capita income, place of residence (urban or rural), self-reported health status, etc; (2) WTP for influenza vaccination; (3) recommended financing sources of influenza vaccination including individuals, medical insurance, and the government; (4) knowledge and perception of influenza and influenza vaccination.
In the original questionnaire, self-reported health status and some variables concerning the knowledge and perception of influenza and influenza vaccination (including perceived possibility of catching influenza, perceived severity of influenza, perceived importance, safety and efficacy of influenza vaccination, and trust in doctors’ vaccination advice) were designed as five-point Likert scales. To simplify the analysis, we regrouped answers “very high” and “high” as “high”, and “fair” ”low” ”very low”as “low” to construct binary variables. We have listed the socio-demographic and health variables recorded in greater detail in the methods section. Please see line 123-135 in the revised manuscript.
- The footnote to Table 3 indicates that *, ** and *** denote significance at the 10, 5 and 1 percent levels, respectively, but 10% is not usually considered statistically significant. A p-value of 0.05 corresponds to an alpha risk of 5%, anything more than that (or >0.05) is not statistically significant by convention. Please comment or correct. If this is a specificity of the statistical methods used, please clarify and explain.
Response 3: Many thanks for Reviewer 1 for pointing out the problem in significance. We agree with your opinion that 10% is not usually considered statistically significant in analysis. We have emphasized in the revised manuscript that a two-sided p-value below 0.05 was considered statistically significant, and changed the results reported in the regression table. Please see line 165-166, 254-282 and Table 4 in the revised manuscript.
- In the references, a review paper might be more appropriate as a reference for the contingent valuation method (e.g. DOI: 10.1016/s0168-8510(99)00010-x), rather than examples of use in individual countries (e.g. refs 11-13).
Response 4: Thanks for your constructive suggestion, which is highly appreciated. We agree with you that a review paper is more appropriate as a reference for the contingent valuation method. We have replaced the several examples of use by a review paper (as recommended, DOI: 10.1016/s0168-8510(99)00010-x). Please see line 138-139 in the revised manuscript.
- Generally speaking, the English warrants thorough revision – there are a number of grammatical mistakes throughout the paper. For example, the authors use the phrase “almost out of pocket” on two different occasions, but there is obviously something missing here – either the expenses are “out-of-pocket”, or they are not. They cannot be “almost” out-of-pocket. There is likely a word missing (e.g. “almost totally out-of-pocket”). Page 11 of 13, line 283 – please correct “serenity” to “severity”. These are only two examples; there are many other parts that warrant correction.
Response 5: Thanks for your kind suggestion, which is valuable for improving the accuracy of the manuscript. We have carefully checked the language throughout, and also invited MDPI language editors to review the manuscript and edit the language. For example, we changed “almost out of pocket” to “totally out-of-pocket on most occasions”, and corrected “serenity” to “severity”. Please see line 63, 293 and 351 in the revised manuscript.
Thanks again to Reviewer 1 for the constructive suggestions on the manuscript. We have studied the comments carefully and have made corresponding corrections. We sincerely hope that this revised manuscript has addressed all your comments and suggestions. Once again, thank you very much for your comments.
Reviewer 2 Report
The manuscript is interesting and original.
Some amendments are required:
- sample size calculations are not given. What is the hypothesis to be tested? Please explain estensively
- what is the meaning of the Tobit coefficients? give a brief description in the Methods section
- in order to make comparisons at the international level, reporting the value of money in US Dollars.
- I would argue that the Peking University Institutional Review Board is an ethical commitee, If so, please clarify.
- some interesting papers concerning the cost-effectiveness of influenza vaccination are missing. Some examples are listed below:
1: Yang J, Atkins KE, Feng L, Baguelin M, Wu P, Yan H, Lau EHY, Wu JT, Liu Y, Cowling BJ, Jit M, Yu H. Cost-effectiveness of introducing national seasonal influenza vaccination for adults aged 60 years and above in mainland China: a modelling analysis. BMC Med. 2020 Apr 14;18(1):90. doi: 10.1186/s12916-020-01545-6. PMID: 32284056; PMCID: PMC7155276.
2: Jiang M, Li P, Wang W, Zhao M, Atif N, Zhu S, Fang Y. Cost-effectiveness of quadrivalent versus trivalent influenza vaccine for elderly population in China. Vaccine. 2020 Jan 29;38(5):1057-1064. doi: 10.1016/j.vaccine.2019.11.045. Epub 2019 Nov 29. PMID: 31787414.
Author Response
Thanks for Reviewer 2 for giving us an opportunity to revise the manuscript.
Responses to Reviewer 2
We would like to express our sincere gratitude to Reviewer 2 for the kind suggestions. We have carefully revised the manuscript according to the constructive suggestions, and the revisions are traced in the revised manuscript. Please find the following detailed point-by-point responses to your comments and suggestions.
- Sample size calculations are not given. What is the hypothesis to be tested? Please explain extensively.
Response 1: Thanks for the notifications from Reviewer 2. We have made corresponding revisions according to your suggestions:
1.1. We added a description of sampling size calculation in the revised manuscript. In the present study, the sample size was calculated under the assumption that the predicted influenza vaccine coverage was 30% among children, 10% among the elderly and 10% among those with chronic diseases. With an allowable error of 5%, a sample size of 323, 138 and 138 for the three groups in each province. To allow for disqualification of incomplete questionnaires, we increased the sample size by 10%, with a final target sample population of 3553, 1518 and 1518 for the three groups in ten provinces. In practice, we collected a larger sample size than expected to increase the reliability of the results. Please see line 96-102 in the revised manuscript.
1.2. The research questions of the present study include examining the WTP and recommended financing sources for seasonal influenza vaccination among children aged 6-59 months, patients with chronic diseases aged 18-59 years old, and the elderly aged above 60 years, trying to find out feasible measures to improve the coverage rate of seasonal influenza vaccination for the three priority groups in China. Based on the research questions, it was hypothesized that the reported WTP was higher among children than the other groups, and a large number of respondents would recommend the government and/or health insurance as financing sources for influenza vaccination. Please see line 74-80 in the revised manuscript.
- What is the meaning of the Tobit coefficients? give a brief description in the Methods section.
Response 2: Thanks for your kind question. In this study, censored WTP values (0–150, concentrate on boundary values) cannot be obtained via the ordinary least squares method, so it is preferable to estimate coefficients using the Tobit model. Results were shown as coefficient (standard error), which can estimate the significance and direction of the exposure effect of independent variables, but cannot directly provide overall exposure effects estimation. The degree of exposure effects can only be obtained after calculating marginal effects.[1] In the revised manuscript, we further obtained the marginal effects of independent variables, and the results were shown in appendix B. Please see line 255-256 and appendix B in the revised manuscript.
For example, table 4 indicates that perceived high possibility of catching influenza would significantly increase WTP among the elderly (significance and direction of the exposure effect), but cannot quantify the exposure effect. The exposure effect can be obtained from marginal effects in appendix B.
- In order to make comparisons at the international level, reporting the value of money in US Dollars.
Response 3: We would like to thank Reviewer 2 for this helpful comment on our manuscript. To make comparisons at the international level, we firstly relocated the equivalence between Yuan and US dollars at the first mention of a price in “Yuan”, and added the equivalence as a note in every table which mentioned “Yuan”. In addition, we have also added the USD values of WTP in the abstract to make convenient comparisons world-wide. Please see line 35-36, 142-143, table 1 and table 2 in the revised manuscript. In the main text, it is recommended to keep the currency of Yuan, so that convenient comparisons can be performed with previously published studies on WTP in China.[2]
- I would argue that the Peking University Institutional Review Board is an ethical committee, If so, please clarify.
Response 4: Yes, Reviewer 2’s argue is correct. Peking University Institutional Review Board is an ethical committee, and we have clarified the fact in the method section, and stated “This study was ethically reviewed and approved by Peking University Institutional Review Board (IRB00001052-19076)”. Please see line 93-94 in the revised manuscript.
- Some interesting papers concerning the cost-effectiveness of influenza vaccination are missing. Some examples are listed below:
1: Yang J, Atkins KE, Feng L, Baguelin M, Wu P, Yan H, Lau EHY, Wu JT, Liu Y, Cowling BJ, Jit M, Yu H. Cost-effectiveness of introducing national seasonal influenza vaccination for adults aged 60 years and above in mainland China: a modelling analysis. BMC Med. 2020 Apr 14;18(1):90. doi: 10.1186/s12916-020-01545-6. PMID: 32284056; PMCID: PMC7155276.
2: Jiang M, Li P, Wang W, Zhao M, Atif N, Zhu S, Fang Y. Cost-effectiveness of quadrivalent versus trivalent influenza vaccine for elderly population in China. Vaccine. 2020 Jan 29;38(5):1057-1064. doi: 10.1016/j.vaccine.2019.11.045. Epub 2019 Nov 29. PMID: 31787414.
Response 5: Thanks for the references suggested by Reviewer 2, which are now included in the revised manuscript. In particular, the recently published study on the cost-effectiveness of introducing influenza vaccination for the elderly in China (the first recommended reference) is interesting and constructive, and it helps a lot in addressing the importance of government financial subsidies and/or medical insurance compensation for influenza vaccination in China. Please see line 297-299 and 333-335 in the revised manuscript.
Thanks again to Reviewer 2 for the kind suggestions on this manuscript. We have studied the comments carefully and have made corresponding corrections. We sincerely hope that this revised manuscript has addressed all your comments and suggestions. The references cited in the response are shown as follows:
- Greene, W. Marginal effects in the censored regression model. Econ Lett 1999, 64, 43–49.
- Hou, Z.; Chang, J.; Yue, D.; Fang, H.; Meng, Q.; Zhang, Y. Determinants of willingness to pay for self-paid vaccines in China. Vaccine 2014, 32, 4471–4477.
Round 2
Reviewer 2 Report
The authors have answered in a satisfactory way to my concerns.
Author Response
Dear Reviewer 2,
We would like to extend our sincere gratitude to you for carefully reviewing our paper and providing helpful comments. Thank you very much for you kind reply!
Best regards